# Cilastatin as a Potential Anti-Inflammatory and Neuroprotective Treatment in the Management of Glaucoma

**DOI:** 10.3390/ijms25063115

**Published:** 2024-03-07

**Authors:** Miguel A. Martínez-López, Sara Rubio-Casado, Diego San Felipe, Beatriz Martín-Sánchez, José A. Fernández-Albarral, Elena Salobrar-García, José A. Matamoros, José M. Ramírez, Rosa de Hoz, Juan J. Salazar, Eva M. Marco, Ana I. Ramírez, Alberto Lázaro, Meritxell López-Gallardo

**Affiliations:** 1Department of Immunology, Ophthalmology and ORL, Faculty of Medicine, Universidad Complutense de Madrid, 28040 Madrid, Spain; miguma19@ucm.es (M.A.M.-L.); ramirezs@med.ucm.es (J.M.R.); 2Department of Physiology, Faculty of Medicine, Universidad Complutense de Madrid, 28040 Madrid, Spain; srubio02@ucm.es (S.R.-C.); diesanfe@ucm.es (D.S.F.); beatrm14@ucm.es (B.M.-S.); alberlaz@ucm.es (A.L.); 3Renal Physiopathology Laboratory, Department of Nephrology, Instituto de Investigación Sanitaria Gregorio Marañón, Hospital General Universitario Gregorio Marañón, 28007 Madrid, Spain; 4Institute of Ophthalmologic Research Ramón Castroviejo, Universidad Complutense de Madrid, 28040 Madrid, Spain; joseaf08@ucm.es (J.A.F.-A.); elenasalobrar@med.ucm.es (E.S.-G.); jomatamo@ucm.es (J.A.M.); rdehoz@med.ucm.es (R.d.H.); jjsalazar@med.ucm.es (J.J.S.); airamirez@med.ucm.es (A.I.R.); 5Department of Immunology, Ophthalmology and ORL, Faculty of Optics and Optometry, Universidad Complutense de Madrid, 28040 Madrid, Spain; 6Health Research Institute, San Carlos Clinical Hospital (IdISSC), 28040 Madrid, Spain; 7Department of Genetics, Microbiology and Physiology, Faculty of Biological Sciences, Universidad Complutense de Madrid, 28040 Madrid, Spain

**Keywords:** retina, glaucoma, ocular hypertension (OHT), retinal ganglion cells (RGCs), macroglial cells, microglial cells, neuroinflammation, cilastatin, therapeutic approach, neuroprotection

## Abstract

Glaucoma is a neurodegenerative disease that causes blindness. In this study, we aimed to evaluate the protective role of cilastatin (CIL), generally used in the treatment of nephropathologies associated with inflammation, in an experimental mouse model based on unilateral (left) laser-induced ocular hypertension (OHT). Male Swiss mice were administered CIL daily (300 mg/kg, i.p.) two days before OHT surgery until sacrifice 3 or 7 days later. Intraocular Pressure (IOP), as well as retinal ganglion cell (RGC) survival, was registered, and the inflammatory responses of macroglial and microglial cells were studied via immunohistochemical techniques. Results from OHT eyes were compared to normotensive contralateral (CONTRA) and naïve control eyes considering nine retinal areas and all retinal layers. OHT successfully increased IOP values in OHT eyes but not in CONTRA eyes; CIL did not affect IOP values. Surgery induced a higher loss of RGCs in OHT eyes than in CONTRA eyes, while CIL attenuated this loss. Similarly, surgery increased macroglial and microglial activation in OHT eyes and to a lesser extent in CONTRA eyes; CIL prevented both macroglial and microglial activation in OHT and CONTRA eyes. Therefore, CIL arises as a potential effective strategy to reduce OHT-associated damage in the retina of experimental mice.

## 1. Introduction

Glaucoma is one of the major leading causes of irreversible blindness worldwide, and its incidence is alarmingly increasing, with the number of glaucomatous patients projected to reach 111.8 million people by 2040 [1,2]. Among glaucomatous patients, primary open-angle glaucoma is most common, and it is characterized by an obstruction of the trabecular pathway that, through the blockade of the conventional pathway of aqueous humor drainage, causes a direct increase in the Intraocular Pressure (IOP) [3]. This abnormal increase in IOP, also called ocular hypertension (OHT), is stated as the most frequent cause of glaucoma [4]. OHT has been reported to exert mechanical pressure over the optic disc, causing the deformation of the retinal ganglion cell (RGC) axons and thus possibly enhancing RGC death due to anterograde and retrograde processes [3,5]. This glaucomatous retinal damage is thus progressive and continuous, and whenever vision loss becomes evident, RGC death is already irreversible [6]. Two main mechanisms have been hypothesized for the etiology of the glaucomatous disease: ischemia–reperfusion processes and retinal inflammatory responses [7,8]. A better understanding of the temporal pattern of the pathophysiological mechanisms underlying glaucoma may open new opportunities for its early diagnosis, as well as for the development of novel therapeutic strategies that may serve as a first step in the management of glaucomatous disease.

In recent decades, animal models have been a great aid in glaucoma research. These have mainly been animal models based upon a direct induction of OHT through surgical interventions such as laser photocoagulation and/or intracameral microbead injection; see [9] for a review. Our research group has broad experience in unilateral laser-induced photocoagulation of the episcleral and limbal veins as an experimental model of glaucoma in mice [4,8]. In this animal model, the IOP peaks one day after surgery, and a gradual and persistent loss of RGCs arises shortly after, starting 3 days after surgery. Remarkably, inflammatory processes involving glial cells are also observed in that time frame, namely, potent microglial activation [10], together with general macroglial activation [11]. It is worth mentioning the changes are observed in the contralateral (CONTRA) normotensive eye. Although milder, activation in glial cells is also evident in the CONTRA eye, and we have reported an increase in microglia cells [10] and an increase in GFAP and MHCII expression [11], as well as a subtle loss of RGCs [8]. Most probably, these changes in the CONTRA eye might be unrelated to direct OHT and associated with induced neuroinflammatory processes.

In the retina, microglial cells are mainly located in the inner and outer plexiform layers and the optic nerve fiber layer–ganglion cell layer (ONFL-GCL) that, in response to neuronal damage, can exert neuroprotective anti-inflammatory action through an M2 phenotype. However, in the case of chronic or uncontrolled damage, the opposite pro-inflammatory M1 microglial phenotype is then observed, leading to neurodegenerative processes [10]. Macroglial cells, including Müller cells and astrocytes, are also activated in response to damage. Müller cells extend across the whole retina, from the inner to the outer limiting membrane, while astrocytes are located in the inner layers of the retina, mainly in the ONFL-GCL [12]. Despite the fact that reactive gliosis may exert early neuroprotective actions, gliosis chronification and/or maladaptive function may induce a disruption in the blood–retinal barrier (BRB), enhancing pro-inflammatory processes; bidirectional communication between macroglial and microglial cells may aggravate these reactive gliosis processes, finally inducing neurodegeneration in RGCs [11,13].

Glaucoma pharmacology is currently based on drugs designed to decrease the IOP that are devoid of secondary adverse effects but are far from effective [3]. Anti-inflammatory compounds have been tested more recently, and herbal extracts such as saffron (*Crocus sativus* L.) [14] and cannabis derivatives [15] have also demonstrated their efficacy. Alternatively, blockades of microglia activation also seem to be of interest, such as caffeine, an adenosine receptor antagonist [16]; ZM241385, an A2AR adenosine receptor antagonist [17], and minocycline [18,19,20,21]. The search for novel neuroprotective strategies may need to focus on additional pathways such metabolism, axon transport, apoptosis, autophagy and/or neuroinflammation processes (see [22] for a review). Herein, we propose cilastatin (CIL) as a potential anti-apoptotic, anti-inflammatory and neuroprotective drug for the management of glaucoma. CIL is a competitive and reversible inhibitor of the dehydropeptidase-I (DHP-I) enzyme, also acting on organic anion transporters (OATs), blocking the internalization of cytotoxic substances and apoptotic, oxidative and inflammatory mediators and preventing cell damage and cell death [23,24,25]. CIL has long been used in human clinical practice in combination with the antibiotic imipenem (imipenem/cilastatin) to prevent hydrolysis of the β-lactam ring of imipenem by enzymes and thus its inactivation, and, in recent years, CIL alone has been shown to prevent renal damage induced by other drug treatments including chemotherapeutics, immunosuppressants and other different antibiotics, among others [23,24,25,26,27]. Actually, these promising results have been recently assayed in patients with peritoneal carcinomatosis undergoing cytoreduction and hyperthermic intraperitoneal intraoperative chemotherapy (HIPEC-cisplatin), where it has been shown that CIL (imipenem/cilastatin) may exert nephroprotective effects [28]. Importantly, a clinical trial is currently underway (EudraCT Number: 2022-001417-39).

Therefore, in the present study, we aimed to evaluate the potential protective role of CIL in a unilateral laser-induced OHT model of glaucoma in male mice. In this study, CIL was administered as a protective therapy before and after induction of OHT, and its efficacy was evaluated (1) for its ability to control IOP levels; (2) for its possible neuroprotective action by quantifying the number of RGCs labeled with the Brn3a antibody; and (3) for its anti-inflammatory capacity, analyzed by studying glial cells, both macroglial and microglial, via immunohistochemical techniques using GFAP and Iba-1 antibodies, respectively. As we have previously characterized the temporal profile of retinal damage in this experimental animal model of glaucoma [10,11,14], days 3 and 7 post-surgery were chosen in this study as the time points of maximum inflammation and neurodegeneration, respectively. Herein, we have also included the normotensive CONTRA eye to better understand the molecular pathways underlying the progression of retinal damage from one eye to the other.

## 2. Results

### 2.1. IOP Measurements

At the basal timepoint, prior to any intervention, normative IOP levels were found in all the animals evaluated (16.16 ± 2.06 mmHg, N = 140, considering that IOP was measured in both eyes from the NAÏVE group); mice were then randomly assigned to different experimental groups.

The results of IOP measurements are shown in Figure 1. In the two cohorts evaluated (animals sacrificed at 3 days, Figure 1A, and 7 days, Figure 1B, after the surgery), the surgery significantly increased IOP levels in OHT eyes compared to both NAÏVE and CONTRA eyes at shorter time points from day 1 to day 3 post-surgery (*p* < 0.001) in the first cohort (Figure 1A) and up to 7 days post-surgery (*p* < 0.001) in the second cohort (Figure 1B). Surprisingly, the IOP in CONTRA eyes was even lower than in NAÏVE eyes not submitted to surgery and mice at longer time points post-surgery (*p* = 0.025). CIL had no effect on IOP levels at any time point analyzed in any of the cohorts investigated (Figure 1).

### 2.2. Analysis of RGCs: Brn3a+ Expression

Results of Brn3a+ expression are shown in Figure 2 and Figure 3 and Appendix A. In the whole retina, three days after the surgery, OHT induced a significant reduction in the number of Brn3a+ cells, as compared to both retinas from NAÏVE and CONTRA eyes; CIL administration did not affect Brn3a expression. Similar results were observed seven days after the surgery: a significant decrease in the number of Brn3a+ cells was observed in response to OHT (OHT + VH vs. NAÏVE + VH, *p* < 0.001), and a more subtle decrease was also observed in the CONTRA eyes (CONTRA + VH vs. NAÏVE + VH, *p* = 0.013). At this time point, CIL was able to prevent and/or revert OHT effects (OHT + CIL vs. OHT + VH, *p* = 0.012).

**Figure 1 ijms-25-03115-f001:**
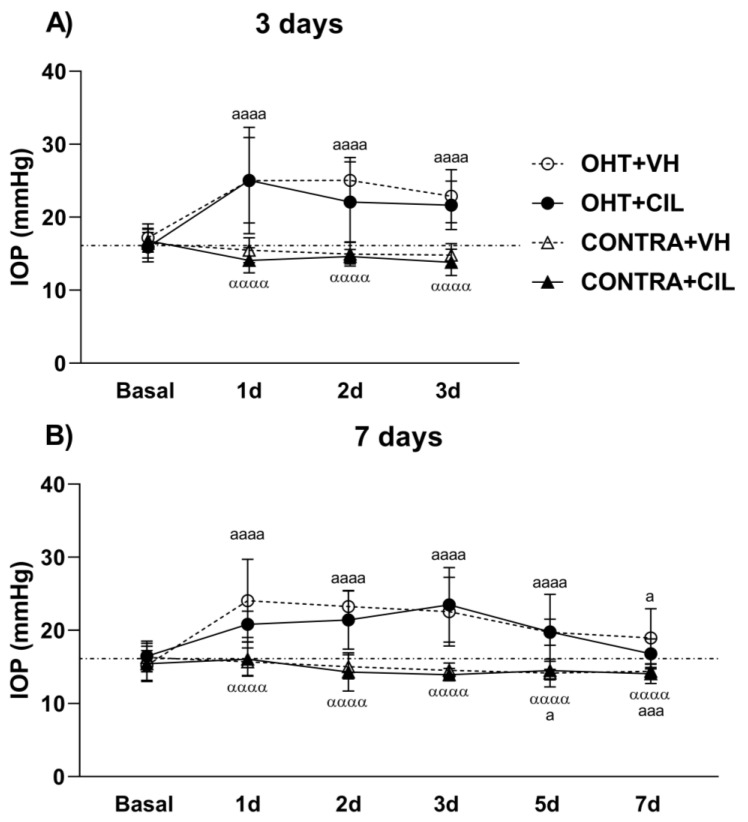
Intraocular Pressure (IOP) measurements. IOP was measured in the left photocoagulated eye (laser-induced ocular hypertension, OHT), and its contralateral (CONTRA) eye. Animals were administered cilastatin (CIL, 300 mg/kg, i.p.) or vehicle (VH, saline) daily two days before the surgery until sacrifice (**A**) 3 or (**B**) 7 days after OHT induction; i.e., animals received 6 or 10 injections, respectively. Experimental groups: OHT + VH (n = 12), OHT + CIL (n = 11), CONTRA + VH (n = 12) and CONTRA + CIL (n = 11) at day 3; OHT + VH (n = 11), OHT + CIL (n = 9), CONTRA + VH (n = 11) and CONTRA + CIL (n = 9) at day 7. IOP levels from the NAÏVE mice, not submitted to surgery, receiving either VH or CIL and sacrificed at both days (n = 44) are represented as a discontinuous line since no drug effect was found. Two-way ANOVA per time point: a and α indicate the post hoc comparisons for the surgery general effect: a *p* < 0.05, aaa *p* < 0.005, aaaa *p* < 0.001 OHT vs. NAÏVE and CONTRA vs. NAÏVE group; αααα *p* < 0.001 CONTRA vs. OHT group.

A further detailed regional analysis of the retina revealed that the decrease in Brn3a+ cells, previously described 3 days after the surgery, was quite homogeneously distributed, as gathered from Figure 2. The reduction in Brn3a+ cells seems to be more abrupt within the temporal retina (for statistical details, consult Appendix A). No effects of the CIL treatment were revealed in this analysis, except for the temporal superior (T-S) area, where the decrease in the OHT + VH group could not be observed in the OHT + CIL mice.

Seven days after the surgery, the reduction in Brn3a+ cells seemed to be notably limited to more nasal and central retinal areas; OHT eyes presented a larger reduction in the number of Brn3a+ cells within the N-C, N-I and C-S areas of the retina. CIL was able to counteract the effects of OHT within these retinal regions, although statistical significance was only achieved in the C-S retina. Instead, the already reported reduction in Brn3a+ cells in the whole retina of CONTRA eyes appeared to reflect a deeper affection of the N-C and N-I regions of the retina (Figure 2; for statistical details, consult Appendix A).

**Figure 2 ijms-25-03115-f002:**
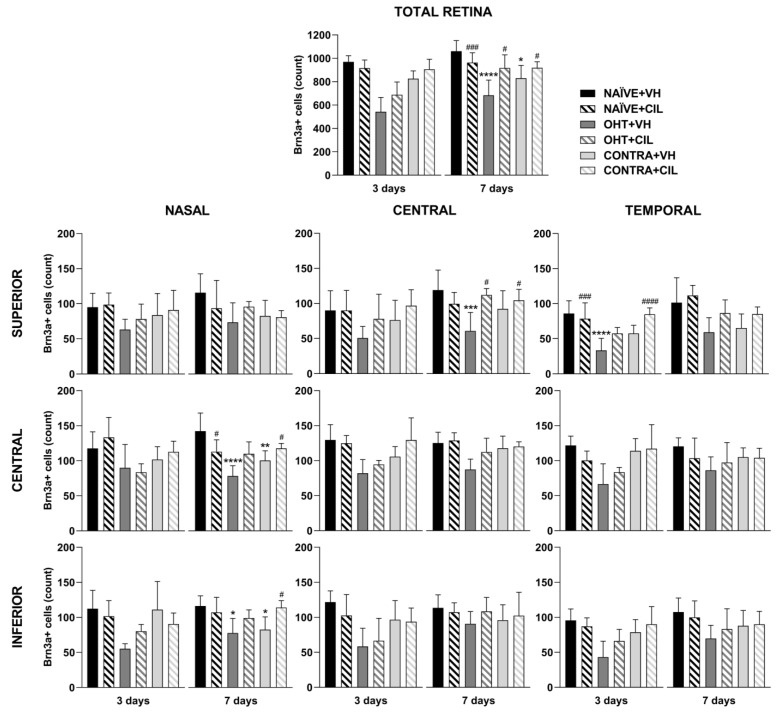
Brn3a+ cell count in the total retina (upper panel) and in the nine retinal areas (lower panel) 3 and 7 days after surgery. Brn3a+ cells were measured in the left photocoagulated eye (laser-induced ocular hypertension, OHT), and its contralateral (CONTRA). Data also include a group of non-manipulated (NAÏVE) mice. Animals were administered cilastatin (CIL, 300 mg/kg, i.p.) or vehicle (VH, saline) daily two days before the surgery until sacrifice 3 or 7 days after OHT induction; i.e., animals received 6 or 10 injections, respectively. Experimental groups: NAÏVE + VH, NAÏVE + CIL, OHT + VH, OHT + CIL, CONTRA + VH and CONTRA + CIL day 3; NAÏVE + VH, NAÏVE + CIL, OHT + VH, OHT + CIL, CONTRA + VH and CONTRA + CIL day 7 (n = 5 per experimental group). Two-way ANOVA followed by Tukey post hoc comparisons in case of a significant interaction between factors. Only significant differences between experimental groups are indicated: * *p* < 0.05, ** *p* < 0.01, *** *p* < 0.005, **** *p* < 0.001 vs. NAÏVE + VH; # *p* < 0.05, ### *p* < 0.005, #### *p* < 0.001 vs. OHT + VH. (See Appendix A for more details from the statistical analysis.) Brn3a: brain-specific homeobox/POU domain protein 3A.

**Figure 3 ijms-25-03115-f003:**
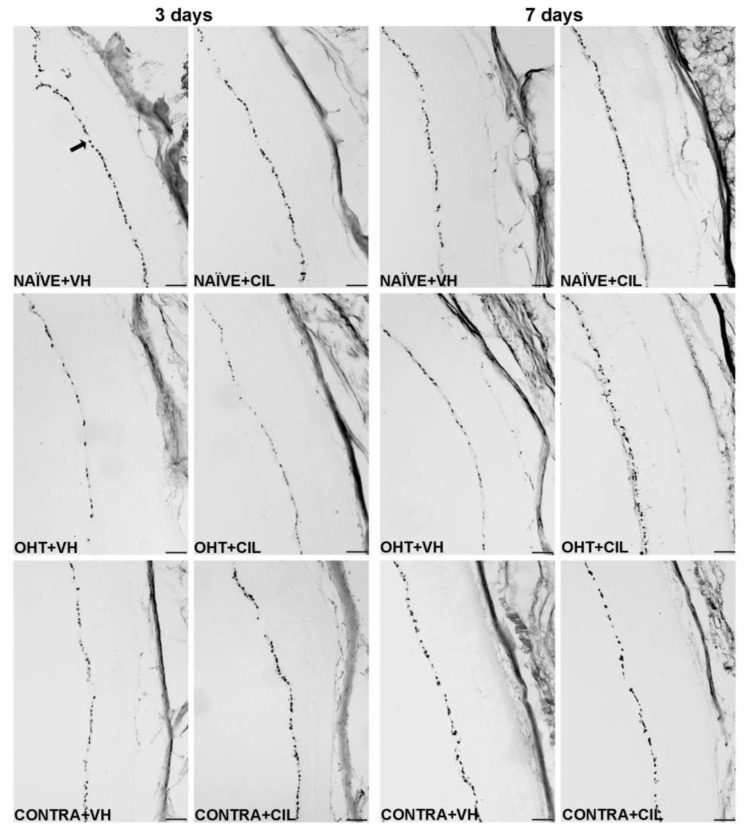
Brn3a + cell count in the total retina 3 and 7 days after surgery. Brn3a+ cells were measured in the left photocoagulated eye (laser-induced ocular hypertension, OHT), and its contralateral (CONTRA). Data also include a group of non-manipulated (NAÏVE) mice. Animals were administered cilastatin (CIL, 300 mg/kg, i.p.) or vehicle (VH, saline) daily two days before the surgery until sacrifice 3 or 7 days after OHT induction; i.e., animals received 6 or 10 injections, respectively. Microphotographs taken at 10× magnification show the main general results of the Brn3a (brain-specific homeobox/POU domain protein 3A) immunohistochemical assay of the different experimental groups 3 and 7 days after OHT induction. In the 3 days NAÏVE + VH microphotograph, an arrow pointing at a Brn3a+ retinal ganglion cell, located in the retinal ganglion cell layer, is included as an example. Scale bar: 50 μm.

### 2.3. Analysis of Macroglial Cells (Astrocytes and Müller Cells): GFAP+ Expression

Results of GFAP+ expression are shown in Figure 4 and Figure 5 and Appendix A. In general, the GFAP expression significantly increased in the whole retina of OHT + VH mice compared to all other groups three days after the OHT induction. The expression of GFAP was significantly increased in the retinas of OHT + VH eyes compared to NAÏVE + VH eyes (*p* = 0.005), and the treatment with CIL prevented and/or reverted this effect (OHT + VH vs. OHT + CIL, *p* = 0.037); the expression of GFAP within the CONTRA eyes did not differ from that observed in control NAÏVE + VH mice. Seven days after OHT induction, a similar pattern was observed, although only general effects were found.

An analysis per retinal subregion (see Figure 4 and Appendix A for statistical details) three days after OHT induction revealed that the OHT-induced increase in GFAP expression was mostly limited to the temporal regions, although also present in the C-I retina. CIL was able to prevent and/or revert these effects, and residual effects of CIL in the N-S region were found, but no differences in GFAP expression were observed in any subregion within the retina of CONTRA eyes (Figure 4). Seven days after OHT induction, the trend of an increase in GFAP expression was observed in all retinal regions, and statistical significance was only achieved in the N-S, T-S and T-I regions. No counteracting effects of CIL were found, and CIL only induced a residual effect, diminishing GFAP expression in the N-I and C-I regions.

**Figure 4 ijms-25-03115-f004:**
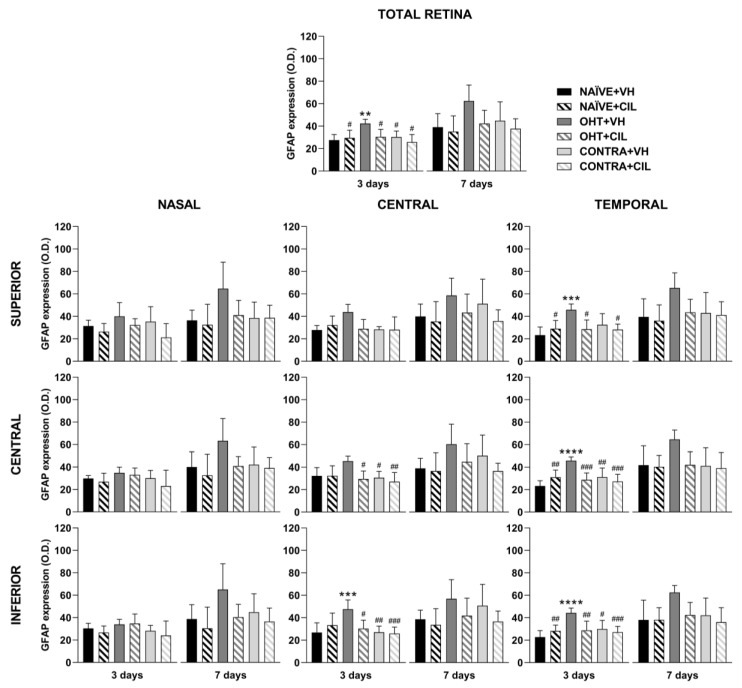
GFAP+ cells optic density (O.D) in the total retina (upper panel) and in the nine retinal areas (lower panel) 3 and 7 days after surgery. GFAP+ cells were measured in the left photocoagulated eye (laser-induced ocular hypertension, OHT), and its contralateral (CONTRA). Data also include a group of non-manipulated (NAÏVE) mice. Animals were administered cilastatin (CIL, 300 mg/kg, i.p.) or vehicle (VH, saline) daily two days before the surgery until sacrifice 3 or 7 days after OHT induction; i.e., animals received 6 or 10 injections, respectively. Experimental groups: NAÏVE + VH, NAÏVE + CIL, OHT + VH, OHT + CIL, CONTRA + VH and CONTRA + CIL day 3; NAÏVE + VH, NAÏVE + CIL, OHT + VH, OHT + CIL, CONTRA + VH and CONTRA + CIL day 7 (n = 5 per experimental group). Two-way ANOVA followed by Tukey post hoc comparisons in case of a significant interaction between factors. Only significant differences between experimental groups are indicated: ** *p* < 0.01, *** *p* < 0.005, **** *p* < 0.001 vs. NAÏVE + VH; # *p* < 0.05, ## *p* < 0.01, ### *p* < 0.005, vs. OHT + VH. (See Appendix A for more details from the statistical analysis.) GFAP: glial fibrillary acidic protein.

**Figure 5 ijms-25-03115-f005:**
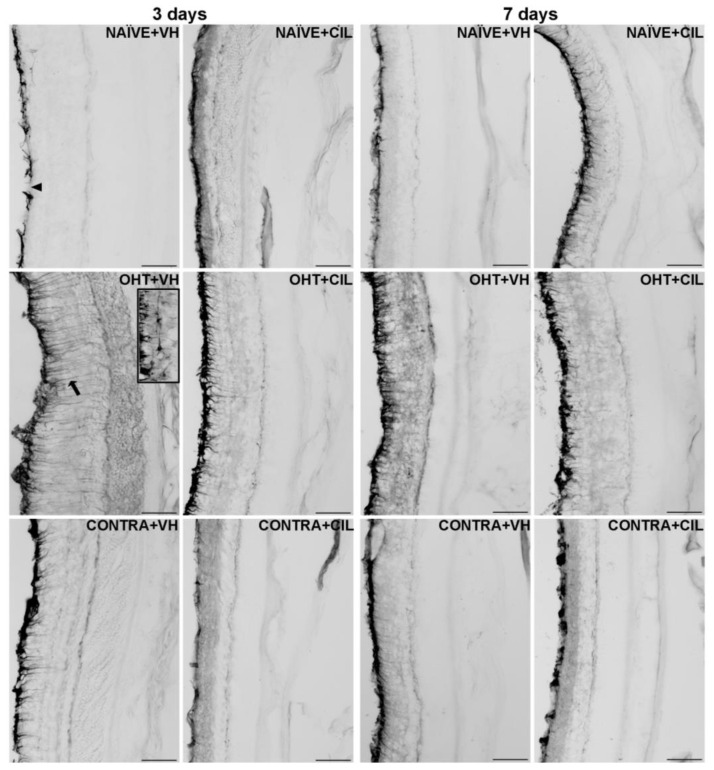
GFAP+ cells optic density (O.D) in the total retina. GFAP+ cells were measured in the left photocoagulated eye (laser-induced ocular hypertension, OHT), and its contralateral (CONTRA). Data also include a group of non-manipulated (NAÏVE) mice. Animals were administered cilastatin (CIL, 300 mg/kg, i.p.) or vehicle (VH, saline) daily two days before the surgery until sacrifice 3 or 7 days after OHT induction; i.e., animals received 6 or 10 injections, respectively. Microphotographs taken at 20× magnification show the main general results of the GFAP (glial fibrillary acidic protein). Immunohistochemical assay of the different experimental groups 3 and 7 days after OHT induction. In the 3 days NAÏVE + VH microphotograph, the optic nerve fiber layer–ganglion cell layer (ONFL-GCL, measured together as a unique layer) is indicated with an arrowhead, and in the 3 days OHT + VH microphotograph, an astrocyte is shown in the insert. The ramifications of a Müller cell are pointed out with an arrow as an example. Scale bar: 50 μm.

GFAP expression was then analyzed across the retinal layers, considering the inner (ONFL/GCL, IPL, INL) and outer (OPL, ONL, PL) layers of the retina separately. In general, an increase in GFAP expression was found only within the inner layers of the retina, with higher levels of GFAP expression in OHT eyes compared to NAÏVE eyes (*p* < 0.05). CIL tended to reduce GFAP expression (see Figure 6 and Appendix A for details). OHT mainly affected the C-S and temporal regions of the inner retina; this increase in GFAP expression also reached the outer layers of the retina in temporal regions (T-S and T-C). As previously indicated, CIL was able to prevent and/or revert the OHT-induced increase in GFAP, even though such an effect seemed to be significant only in the inner layers of the retina. Changes observed in the outer retina in OHT + VH animals were not present in the OHT + CIL-treated groups (see Figure 6 and Appendix A for statistical details). Seven days after the surgery, an increase in the GFAP expression in the inner retinal N-S, N-I, T-S and T-I regions of OHT animals was observed, together with a general downregulation of GFAP expression due to the pharmacological treatment with CIL in the N-C, N-I and C-C inner retinal regions. This same effect was observed in the N-I, C-I, T-C and T-I outer retinal regions (see Figure 6 and Appendix A for statistical details).

### 2.4. Analysis of Microglial Cells: Iba-1+ Expression

Results of Iba1+ expression are shown in Figure 7 and Figure 8 and Appendix A. Three days after the surgery, we found that OHT + VH retinas expressed a significantly higher number of Iba-1+ cells compared to NAÏVE + VH eyes (*p* < 0.001), and CONTRA + VH eyes also exhibited an increased number of Iba-1+ cells than NAÏVE + VH eyes (*p* < 0.001). CIL administration was able to prevent and/or revert such an increase (OHT + CIL vs. OHT + VH eyes, *p* < 0.001; CONTRA + CIL vs. CONTRA + VH eyes, *p* < 0.001). Similar results were found seven days after OHT induction: OHT + VH eyes exhibited a significantly higher number of Iba-1+ cells than NAÏVE + VH eyes (*p* < 0.001), CONTRA + VH eyes also showed a significant increase, although milder, in the number of Iba-1+ cells compared to NAÏVE + VH eyes (*p* < 0.001) and, in both cases, CIL was able to prevent and/or revert such effects (OHT + CIL vs. OHT + VH, *p* < 0.001; CONTRA + CIL vs. CONTRA + VH, *p* < 0.001).

In the further regional analysis, three days after OHT induction, we observed that the number of Iba-1+ cells was significantly and remarkably increased in all retinal regions when compared to NAÏVE + VH eyes. CONTRA + VH eyes also showed a significant although milder effect on the number of Iba-1+ cells; i.e., there was an increase in some nasal and central regions of the retina, but not in temporal regions. Samples receiving CIL did not exhibit a surgery-induced increase in Iba-1+ cells, as was the case for OHT eyes and CONTRA eyes (see Figure 7 and Appendix A for statistical details). Seven days after OHT induction, the increase in the amount of Iba-1+ cells was still present in OHT + VH eyes in all retinal regions, and the increase in the number of Iba-1+ cells in CONTRA + VH eyes was restricted to the C-C and T-S regions; in all cases, such an increase in the number of Iba-1+ cells was blocked and/or attenuated by the pharmacological treatment with CIL (see Figure 7 and Appendix A for statistical details). Results with non-relevant biological information are not shown.

The distribution of Iba-1+ cells within specific retinal layers was also investigated, and these microglial cells in OHT eyes appeared to be mainly detected within the ONFL-GCL, IPL, OPL and ONL-PL groups, both at three and seven days after the surgery. In CONTRA + VH eyes, the increase in Iba-1+ cells was only detected in IPL and OPL layers at three days after the surgery and exclusively in the IPL at seven days after the surgery. CIL administration avoided this surgery-induced increment in Iba-1+ cells in all these retinal layers, both in OHT and CONTRA eyes (Figure 9, consult Appendix A for statistical details). Results with non-relevant biological information are not shown. Microglial morphotypes were also investigated, and RD microglial cells were detected as the majority, followed by HZ microglial cells, with a low percentage of VT cells. No differences in this pattern of microglial morphotypes were observed due to the surgical intervention or due to CIL treatment at any time point.

**Figure 6 ijms-25-03115-f006:**
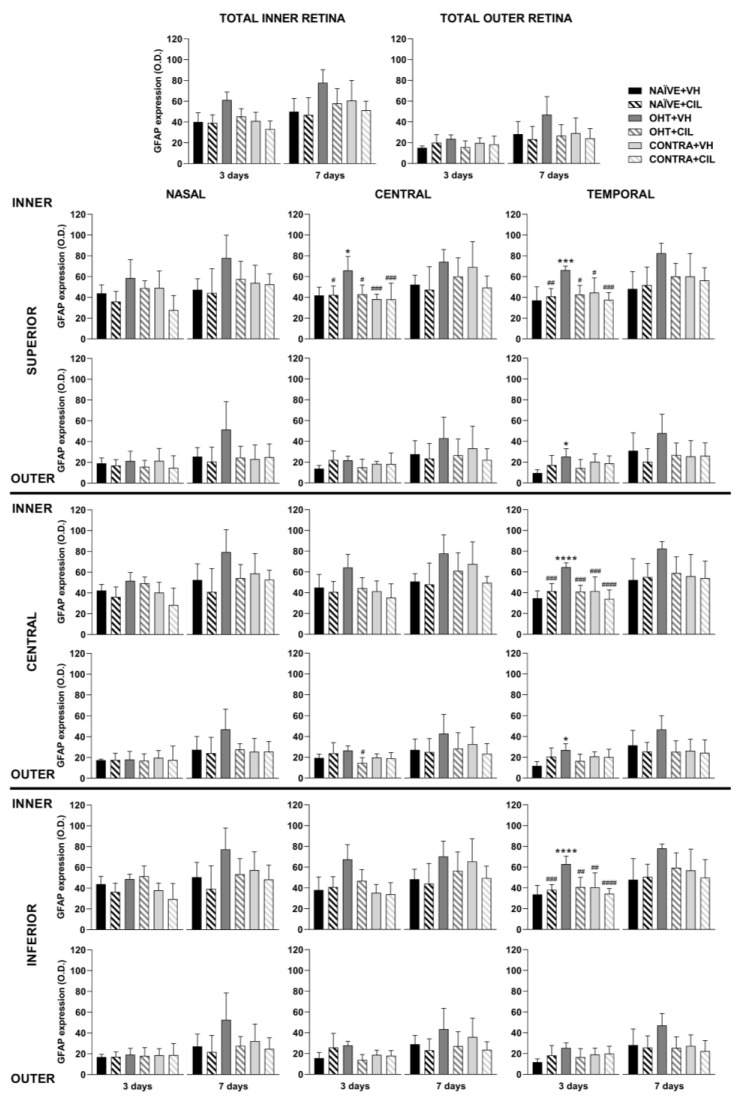
GFAP+ cells optic density (O.D.) in the total retina (upper panel) and in the nine retinal areas (lower panel) 3 and 7 days after surgery. GFAP+ cells were measured in the left photocoagulated eye (laser-induced ocular hypertension, OHT), and its contralateral (CONTRA). Data also include a group of non-manipulated (NAÏVE) mice. Animals were administered cilastatin (CIL, 300 mg/kg, i.p.) or vehicle (VH, saline) daily two days before the surgery until sacrifice 3 or 7 days after OHT induction; i.e., animals received 6 or 10 injections, respectively. Experimental groups: NAÏVE + VH, NAÏVE + CIL, OHT + VH, OHT + CIL, CONTRA + VH and CONTRA + CIL day 3; NAÏVE + VH, NAÏVE + CIL, OHT + VH, OHT + CIL, CONTRA + VH and CONTRA + CIL day 7 (n = 5 per experimental group). Two-way ANOVA followed by Tukey post hoc comparisons in case of a significant interaction between factors. Only significant differences between experimental groups are indicated: * *p* < 0.05, *** *p* < 0.005, **** *p* < 0.001 vs. NAÏVE + VH; # *p* < 0.05, ## *p* < 0.01, ### *p* < 0.005, #### *p* < 0.001 vs. OHT + VH. (See Appendix A for more details from the statistical analysis.) GFAP: glial fibrillary acidic protein.

**Figure 7 ijms-25-03115-f007:**
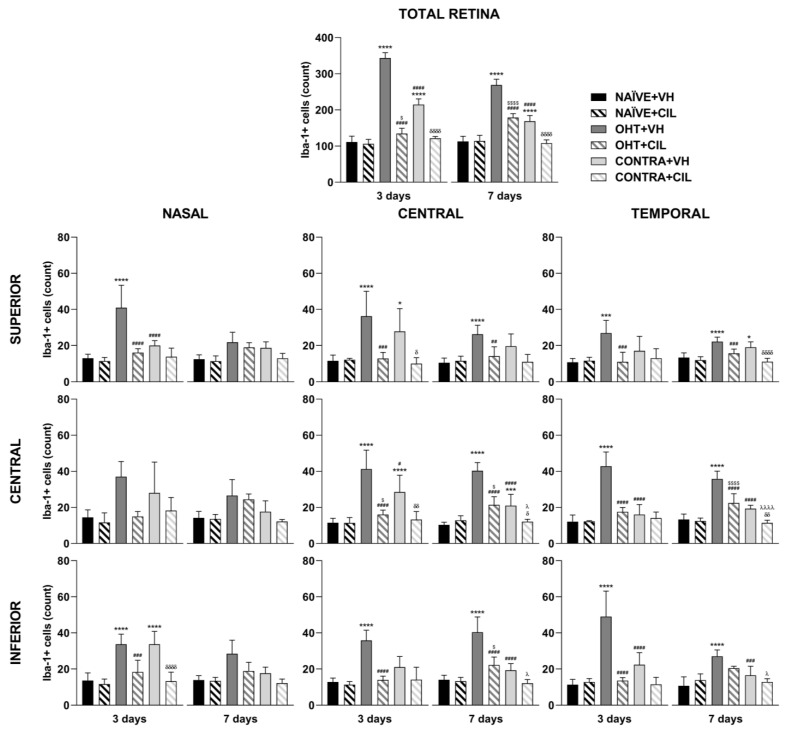
Iba-1+ cell count in the total retina (upper panel) and in the nine retinal areas (lower panel) 3 and 7 days after surgery. Iba-1+ cells were measured in the left photocoagulated eye (laser-induced ocular hypertension, OHT), and its contralateral (CONTRA). Data also include a group of non-manipulated (NAÏVE) mice. Animals were administered cilastatin (CIL, 300 mg/kg, i.p.) or vehicle (VH, saline) daily two days before the surgery until sacrifice 3 or 7 days after OHT induction; i.e., animals received 6 or 10 injections, respectively. Experimental groups: NAÏVE + VH, NAÏVE + CIL, OHT + VH, OHT + CIL, CONTRA + VH and CONTRA + CIL day 3; NAÏVE + VH, NAÏVE + CIL, OHT + VH, OHT + CIL, CONTRA + VH and CONTRA + CIL day 7 (n = 5 per experimental group). Two-way ANOVA followed by Tukey post hoc comparisons in case of a significant interaction between factors. Only biologically relevant and significant differences between experimental groups are indicated: * *p* < 0.05, *** *p* < 0.005, **** *p* < 0.001 vs. NAÏVE + VH; # *p* < 0.05, ## *p* < 0.01, ### *p* < 0.005, #### *p* < 0.001 vs. OHT + VH; $ *p* < 0.05, $$$$ *p* < 0.001 vs. NAÏVE + CIL; δ *p* < 0.05, δδ *p* < 0.01, δδδδ *p* < 0.001 vs. CONTRA + VH; λ *p* < 0.05, λλλλ *p* < 0.001 vs. OHT + CIL. (See Appendix A for more details from the statistical analysis.) Iba-1: ionized calcium-binding adapter molecule 1.

**Figure 8 ijms-25-03115-f008:**
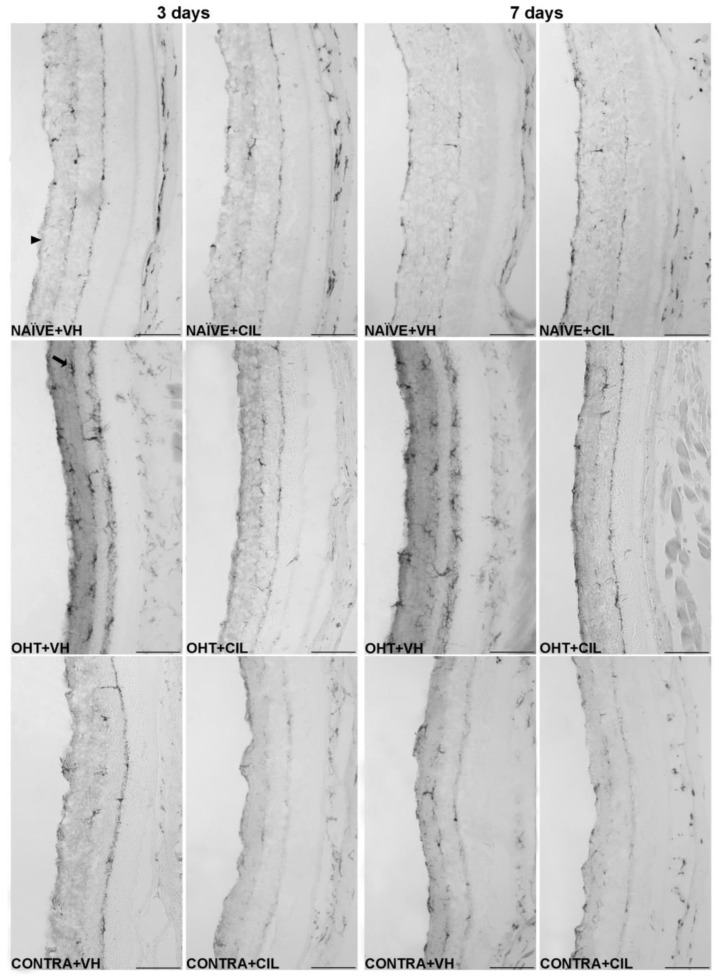
Iba-1+ cells in the total retina. Iba-1+ cells were measured in the left photocoagulated eye (laser-induced ocular hypertension, OHT), and its contralateral (CONTRA). Data also include a group of non-manipulated (NAÏVE) mice. Animals were administered cilastatin (CIL, 300 mg/kg, i.p.) or vehicle (VH, saline) daily two days before the surgery until sacrifice 3 or 7 days after OHT induction; i.e., animals received 6 or 10 injections, respectively. Microphotographs taken at 20× magnification show the main general results of the Iba-1 (ionized calcium-binding adapter molecule 1) immunohistochemical assay of the different experimental groups 3 and 7 days after OHT induction, where the main effects can be appreciated in the graph. In the 3 days NAÏVE + VH microphotograph, the arrowhead points to the optic nerve fiber layer–retinal ganglion cell layer. In the 3 days OHT + VH image, the arrow points to an example of a microglial cell. Scale bar: 50 μm.

**Figure 9 ijms-25-03115-f009:**
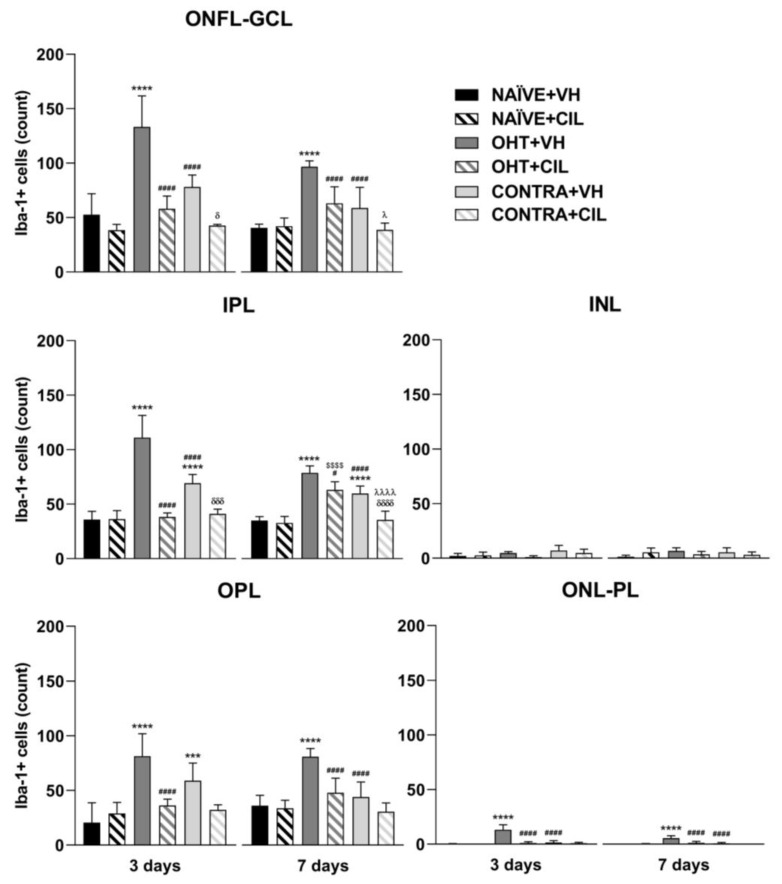
Iba-1+ cell count in the different retinal layers of mice 3 and 7 days after surgery. Iba-1+ cells in the optic nerve fiber layer–ganglion cell layer (ONFL-GCL, measured together as an unique layer), inner plexiform layer (IPL), inner nuclear layer (INL), outer plexiform layer (OPL) and outer nuclear layer–photoreceptor layer (ONL-PL, measured together as an unique layer). Iba-1+ cells were measured in the left photocoagulated eye (laser-induced ocular hypertension, OHT), and its contralateral (CONTRA). Data also include a group of non-manipulated (NAÏVE) mice. Animals were administered cilastatin (CIL, 300 mg/kg, i.p.) or vehicle (VH, saline) daily two days before the surgery until sacrifice 3 or 7 days after OHT induction; i.e., animals received 6 or 10 injections, respectively. Experimental groups: NAÏVE + VH, NAÏVE + CIL, OHT + VH, OHT + CIL, CONTRA + VH and CONTRA + CIL day 3; NAÏVE + VH, NAÏVE + CIL, OHT + VH, OHT + CIL, CONTRA + VH and CONTRA + CIL day 7 (n = 5 per experimental group). Two-way ANOVA followed by Tukey post hoc comparisons in case of a significant interaction between factors. Only significant differences between experimental groups are indicated: *** *p* < 0.005, **** *p* < 0.001 vs. NAÏVE + VH; # *p* < 0.05, #### *p* < 0.001 vs. OHT + VH; $$$$ *p* < 0.001 vs. NAÏVE + CIL; δ *p* < 0.05, δδδ *p* < 0.005, δδδδ *p* < 0.001 vs. CONTRA + VH; λ *p* < 0.05, λλλλ *p* < 0.001 vs. OHT + CIL. (See Appendix A for more details from the statistical analysis.) Iba-1: ionized calcium-binding adapter molecule 1.

## 3. Discussion

In the present study, we have successfully applied an experimental animal model of glaucoma via unilateral laser-induced OHT in mice, as in previous studies from our research group [8,10,11,14]. We have effectively induced a transient increase in IOP levels exclusively in the left eye, reaching a peak three days after surgery that returned to normal levels seven days after. In the OHT eyes, a decrease in the total number of Brn3a+ cells, which are RCGs, was observed at the IOP peak, probably directly related to the mechanical damage induced by the increase in the IOP as previously suggested [29]. However, some alternative underlying mechanisms might also be proposed and are discussed in this section, mainly processes related to glial activation. In this regard, we deepened the understanding of the pathogenesis of glaucoma by further analyzing the inflammatory retinal processes in both different anatomical retinal areas and layers.

### 3.1. IOP

As previously stated, by performing unilateral OHT induction surgery, temporal OHT was achieved in the left (OHT) eye, demonstrated by increased IOP values. The untreated right CONTRA normotensive eye did not exhibit any changes in IOP measurements.

In this study, we have analyzed the possible protective effects of CIL in the glaucomatous pathology, but we found no significant effects of this drug on the measured IOP values; however, it is worth mentioning that a reduction in the IOP is not the only factor required for a neuroprotective effect in glaucoma. Actually, in previous studies, we have demonstrated that saffron is an antioxidant and a neuroprotective agent in this experimental animal model of glaucoma, despite not reducing the laser-induced increase in IOP [14]. Moreover, we cannot exclude that the lack of effects of these drugs on IOP values might be related to the fact that, in our animal model, we artificially induced OHT via photocoagulation of the limbal and episcleral veins, a mechanism that is completely different from the one that occurs in the clinic in primary open-angle glaucoma.

### 3.2. Retinal Ganglion Cells

A similar, although milder, decrease in RGCs was still observed seven days after the surgery. Previous studies have also reported a progressive reduction in Brn3a+ cells following OHT surgery in a comparable time window, further suggesting a major effect in temporal and central retinal regions, with the nasal regions being less affected at short time points (3 and 5 days after the surgery), and the inferior zones being barely affected at extended time points (7 and 15 days after the surgery) [8]. Moreover, additional studies have reported a long-term downregulation of Brn3a staining, even one month after OHT induction, with a similar regional pattern of RGC loss [29]. In parallel, the present detailed retinal analysis by region allowed us to identify that, three days after surgery, the RGC loss was generalized, although temporal regions seemed to be mostly affected. In our study, Brn3a staining seems to recover partially from the OHT-induced damage seven days after the surgery, since the effects on Brn3a expression reported at this time point were subtler and limited to some nasal and central regions. In this regard, it is worth mentioning previous results from our group, in which a transitory loss of Brn3a expression was reported between days 5 and 7 after OHT induction. This downregulation was probably caused by the inflammatory damage that directly affects RGCs [8]. The mechanisms underlying changes in Brn3a expression in the glaucomatous retina need to be further investigated. The previous literature has highlighted that Brn3a might not be essential for adult RGC survival in normal and damaged retinas [30], and that Brn3a staining may still be present in non-functionally competent RGCs with retrograde axonal transport impairments [29]. Thus, in the future, our studies may not only consider counting studies of Brn3a-stained cells but may also include complementary analyses, such as an analysis of RGC cell size and distribution [31] as well as RCG functionality [29].

As previously indicated, the CONTRA eye is normotensive in this experimental animal model, although it seems to be considerably affected [22]. By analyzing Brn3a+ cells, the CONTRA eyes were shown to be affected in a less significant way at both time points. Contrary to the results observed in OHT eyes, the most affected areas seem to be the ones closer to the optic disc. In this regard, despite some studies reporting no affection of Brn3a in the CONTRA eye [29], some others have reported a slight decrease in the CONTRA eyes in some temporal areas [8]. These effects in the CONTRA eye have been related to the glial-induced cytokine dysregulation that peaks within this time window [8], probably due to the inflammatory signals that travel through the optic nerve of the OHT eye and reach the optic nerve of the CONTRA eye, further traveling anterogradely to the retina [32].

In the current study, we have analyzed the possible protective effects of CIL in the glaucomatous pathology. CIL returned the number of Brn3a+ cells in the OHT + CIL eyes to values closer to the NAÏVE ones mainly in the T-S area by day 3, and in N-C, N-I and C-S areas by day 7. CIL was also able to revert all the changes found in CONTRA eyes.

### 3.3. Macroglial Cells

Regarding glial activation, remarkable increases in GFAP expression were observed both three and seven days after OHT induction. Left OHT eyes exhibited a higher GFAP expression three days after the surgery, with a major impact in the temporal retina, where an augmented GFAP expression was not only present in the inner retina (ONFL-GCL, IPL and INL) but also reached the outer retina (OPL, ONL and PL), suggesting an acute activation of Müller cells that mainly extended their active phenotype across all retinal layers up to the PL. Indeed, a previous study has already described the presence of reactive Müller glia across the retina of OHT eyes, as well as their end feet along the retinal surface, at similar time points after surgery [11]. The observed increment in GFAP expression could also indicate an activation of astrocytes; indeed, a previous study has already reported that astrocytes have thicker somas and more secondary processes in OHT eyes compared to naïve eyes at similar time points after laser treatment, highlighting a region-specific effect in the peripheral and intermediate retina. Perhaps this higher activation of GFAP in the temporal/peripheral retina could be explained by the increased ability of these astrocytes to respond to environmental injuries in order to provide a higher level of protection in the limiting transitional area of the retina near to the ciliary body, as previously proposed [11]. Surprisingly, OHT did not affect GFAP expression in nasal regions, that is, the retinal area nearer to the optic nerve head, in accordance with previous results from our group [11]. Researchers have suggested that these central astrocytes, located in the surroundings of the optic nerve head, exhibit different metabolic capacities and a reduced ability to maintain BRB integrity, possibly due to a downregulation of GFAP expression that has been observed in other models of glaucoma [33,34]. It is worth mentioning that, in our case, this OHT-induced increment in GFAP expression was not so remarkable seven days after the surgery, even though it could still be observed in some temporal regions within the inner retina.

In this study, we did not find any difference in GFAP expression in CONTRA eyes compared to NAÏVE ones, in contrast to what we have previously found in our studies [11]. These differences could be due to the different methodological approaches used in each study, since in the current study, we analyzed retinal sections, while in the previous one, we used whole-mounted retinas. Another study that analyzed the effects of glaucoma in rats found alterations in the astrocytes and Müller cells in the CONTRA eye 1 month after surgery [33], so maybe if we extended our study further in time, we would observe macroglial activation in the CONTRA eyes.

CIL decreased GFAP expression at both time points after the surgery, attenuating the damage in the most affected areas of OHT eyes. The OHT-induced increase in GFAP activation within the inner and outer retina was no longer observed after CIL administration in any area, seemingly affecting the whole retina, even reaching the less irrigated and most affected temporal regions.

### 3.4. Microglial Cells

Microglial activation, detected as an increase in Iba-1 expression, was robust and generalized in all retinal regions at the two time points considered. Microglial activation includes cell proliferation, cell migration and changes in cell morphology and phenotype towards microglia with a higher soma and retracted prolongations with phagocytic properties [10,35]. In particular, we observed a highly significant increase in the number of Iba-1+ cells due to OHT, together with a migration of these microglial cells towards deeper outer layers, reaching the INL and OPL three days after surgery and the ONL-PL both at three and seven days after surgery. We observed an increase in the number of Iba-1+ cells in OHT + VH eyes in each layer, with the most prominent changes in the ONFL-GCL, concluding that this may be the most affected layer of the retina due to the presence of damaged RGCs, especially at day 3. However, the morphological analysis of microglial morphotypes did not show any difference due to the OHT-induced injury, with most microglial cells showing a surveillance (RD) phenotype.

It is worth highlighting that the peak of microglial activation, three days after OHT induction, as already described [10], coincides with the peak in IOP, as well as with the highest point of macroglial activation and RGC degeneration. However, the former event (microglial activation) seemed to be generalized all over the retina, whereas the latter events (macroglial activation and RGC degeneration) were more restricted to the temporal regions of the retina. A possible explanation for these observations might lie in the direct consequences of the increment in IOP, degenerating RGCs and, thus, activating the macroglial response, where both astrocytes and Müller cells might play a role in a potential neuroprotective mechanism. In parallel, microglial cells may activate and mobilize in response to this diffuse alteration, maybe in response to the cytokine and chemokine dysregulation already observed at shorter times [8,36].

CONTRA eyes had milder but similar microglial alterations compared to OHT eyes, mainly being affected in the areas of the retina closer to the optic disc (nasal and central) 3 days after OHT induction, with more moderate changes at day 7. We already observed changes in the CONTRA eyes in our previous study [10] although this was not paired with RGC degeneration; thus, we hypothesize that it could be due to a disruption in the BRB due to OHT-induced damage, which may induce the release of inflammatory signals through the bloodstream to the CONTRA eye. Herein, we propose that signaling through the optic nerve of the OHT eye may also explain the higher microglial activation in the nasal and central areas of the CONTRA eye, as previously reported in a similar model of unilateral ocular damage [32]. Regarding the analysis per retinal layer, the CONTRA eyes only differed from the NAÏVE eyes in both plexiform layers, with these being the two retinal layers with the highest amount of microglial cells, probably having a higher capability to respond to retinal damage than the other layers.

### 3.5. Cilastatin

CIL seemed to attenuate the effects of OHT induction on microglial cells in both OHT and CONTRA eyes at both time points. In further regional and layer analyses, CIL seemed to revert most of these changes, suggesting anti-inflammatory effects, diminishing the microglial response in the retina. Even though it had a generalized response over all the retina, the effects appear to be more remarkable in areas closer to the optic disc, returning the microglial activation to values near to those observed in NAÏVE animals in both eyes.

As previously explained, CIL is a competitive and reversible inhibitor of DHP-I, a glycoprotein located in the cholesterol lipid rafts of the renal epithelial brush border that hydrolyzes peptides. These lipid rafts participate in transmembrane transportation, endocytosis and signaling. CIL can also selectively inhibit OATs, which actively transport substances like drugs from the blood into the cells. CIL preserves renal tubular epithelial cells from the deterioration caused by nephrotoxicity, since the inhibition of OATs and DHP-1 also inhibits the endocytosis that takes place in lipid rafts, avoiding the accumulation of nephrotoxic agents in renal cells, inhibiting inflammatory processes and reducing ROS production and apoptosis [23,24,25,27,37]. There is no research in the literature reporting this, but in the current study, since it was administered intraperitoneally, we believe that CIL or its derived metabolites have the potential to pass the BRB and affect specific receptors in retinal cell populations, probably similar ones to those activated in the kidney. We cannot yet assure whether CIL may directly affect RGCs or attenuate the activation of glial cells, reducing the local inflammatory response of the retina due to an inhibition of internalization of the inflammatory and apoptotic mediators (and therefore their signaling pathways) released by glial cells through the blockage of membrane lipid rafts and OATs. It is well known that microglial and macroglial cells bidirectionally communicate, activating each population reciprocally [13,38]; thus, if CIL is blocking microgliosis, macrogliosis or both, it will contribute to a moderation in the retinal inflammatory response and further RGC death. Even though CIL affected the whole retina, the treatment seemed to be most effective in the regions closer to the optic disc. Protecting this disc is crucial, since in most mammals (not rodents), the fovea, which is the area with the highest visual acuity, is in the central retina, proximate to the optic disc [39].

### 3.6. Limitations

In the present study, only male mice were used, and it could be useful to replicate the study using female mice. Even though most affected glaucomatous patients are male, it has been demonstrated that women of a fertile age taking contraceptives have increased chances of developing glaucoma. We also need to deepen our comprehension of the molecular mechanisms underlying the effects of CIL in the treatment of glaucoma. This study was performed only on days 3 and 7 because, in previous work in our group, we showed that there was either more inflammation and more glial activation or more damage in the ganglion cell population. It would be interesting to continue analyzing the temporal progression of the damage and the effects of the drug over time.

## 4. Materials and Methods

This study was designed and performed in accordance with the European Directive 2010/63/EU, in compliance with the Spanish Royal Decree 53/2013 for the protection of animals used for research and other scientific purposes, and the Association for Research in Vision and Ophthalmology (ARVO) statement for the use of animals in ophthalmic and vision research. This research project was approved by the Local Ethics Committee on Animal Experimentation of the Complutense University of Madrid (CEEA-UCM, 17 July 2023), and authorized by the Approval Board of the Madrid Region Government (Comu-nidad de Madrid, PROEX: 158.1/23). In addition, this work was written following the ARRIVE guidelines (Animal Research: Reporting of In Vivo Experiments), which contain recommendations for describing and publishing studies carried out using animals for experimental purposes [40].

### 4.1. Animals

A total of 72 male albino Swiss mice were purchased from Charles River Laboratories (Les Oncins, France). Mice arrived at the animal facilities in the School of Medicine of the Complutense University of Madrid (ANUC ES28079000086) with a mean weight of 36.81 ± 1.29 g. Animals were housed in pairs in polycarbonate cages (55 cm× 32 cm × 18 cm), with cotton and cardboard tubes as environmental enrichment, and were maintained under controlled environmental conditions (temperature, 21 ± 1 °C and humidity, 60 ± 10%) in a 12 h light–dark cycle (lights on at 7:00) with free access to water and pelleted food (Altromin^®^, Lage, Germany). On arrival, mice were allowed one week of habituation in which animals were not manipulated.

### 4.2. Drug Administration

CIL [sodium (Z)-7-[[(2R)-2-amino-2- carboxyethyl] sulfanyl]-2-[[[(1 S)-2,2-dimethyl cyclopropyl] carbonyl] amino ]- hept-2-enoate] (Sun Pharmaceutical Industries Limited, Madhya Pradesh, India) powder was stored at 4 °C and dissolved in saline (0.9% NaCl) at a concentration of 60 mg/mL. Solutions were made daily. Mice were daily administered the drug intraperitoneally (i.p.) at a dose of 300 mg/kg according to [41] 2 days before the surgery, immediately after the OHT laser induction, and thereafter daily until sacrifice (between 12:00 and 15:00). Depending upon the day of sacrifice, either 3 or 7 days after the surgery, animals received 6 or 10 injections, respectively. Control animals received the same number of saline injections [CIL vehicle (VH), 5 mL/kg, i.p] as their counterparts.

### 4.3. Experimental Animal Model of Glaucoma

As described in [4], surgery was performed on anesthetized mice to induce ocular hypertension (OHT). An i.p. mixture of medetomidine (0.26 mg/kg; Medetor^®^, Virbac España S.A., Barcelona, Spain) and ketamine (75 mg/kg; Anesketin^®^, Dechra Veterinary Products SLU, Barcelona, Spain) was employed to anesthetize the mice. The left eye of the mice was treated with a diode laser beam, applied without any lens, on the episcleral and limbal veins; these veins were cauterized by performing 55–76 burns with a spot size of 50–100 µm for 0.5 s at a power of 0.3 W. After the surgery, tobramycin (Tobrex^®^; Alcon, Barcelona, Spain) was topically applied to prevent eye drying and infection. Animals rapidly recovered via the subcutaneous administration of atipamezole hydrochloride (0.013 mg/g; Antidorm^®^, Calier, Barcelona, Spain), and additional eye drops (sodium hyaluronate (VisuXL, VISUfarma, Madrid, Spain) were applied to both eyes. Animals were observed and kept warm until total recovery.

### 4.4. Intraocular Pressure (IOP) Measurements

Intraocular Pressure (IOP) was measured as previously described [14] by using a rebound tonometer (Tono-Lab, Tiolat, OY, Helsinki, Finland) in anesthetized mice (2% isoflurane, ISOFLO ^®^, Zoetis SL, Madrid, Spain, under an oxygen flow of 1.5 L/min). The IOP was measured in both eyes of animals from all experimental groups. Each IOP value was obtained from 3 independent measurements, each 1 resulting from the automatic mean of 6 consecutive measures. IOP was measured at the beginning of the experiment, prior to the first drug administration, 2 days before surgery (basal IOP) and at five times after OHT induction surgery: 1, 2, 3, 5 and 7 days. IOP measurements were always taken at the same day time (10:00) to avoid possible variations due to circadian rhythms.

### 4.5. Experimental Design

Seventy-two animals (N = 72) were randomly distributed into different experimental groups (see Table 1 for details), and no differences in the initial body weight were observed between groups: 36.6 ± 1.38 g (NAÏVE + VH); 37.48 ± 1.04 g (NAÏVE + CIL); 36.85 ± 1.25 g (OHT + VH) and 36.54 ± 1.38 g (OHT + CIL).

### 4.6. Tissue Preparation

Animals were sacrificed with an overdose of medetomidine (Medetor^®^, Virbac España S.A., Barcelona, Spain) and ketamine (Anesketin^®^, Dechra Veterinary Products SLU, Barcelona, Spain). Mice were transcardially perfused with saline solution (0.9% NaCl), and then with a 4% paraformaldehyde (PFA) solution in 0.1 M phosphate buffer using a volume of 1500 mL/g of bodyweight for both saline and PFA, and both solutions at 4 °C. Eyes were extracted and a previous suture point in the upper eyelid was used to enable the spatial orientation of the retina. Eyes were immersed in 4% PFA at 4 °C for 24 h; then, corneas and lenses were removed and the optical cups containing the retina were left in 4% PFA at 4 °C for 24 h. Then, each eye was washed 3 times for 30 min each time in a phosphate saline buffer solution (PBS, pH 7.2). Thereafter, for cryoprotection purposes, eyes were embedded in sucrose solutions of increasing concentrations. First, eyes were left in an 11% sucrose solution in PBS for 24 h at 4 °C, and then, the solution was changed to a 33% sucrose solution in PBS at 4 °C for an additional 48 h. Lastly, the eyes were placed in a tissue-freezing medium (Tissue-Tek^®^ O.C.T. Compound, Sakura Finetek, Barcelona, Spain). Eye anatomical references were tracked throughout the sample processing, and eye samples were then stored at −30 °C until use [14].

For the immunohistochemical analysis, optical cups with their retinas were frozen and sectioned into 16 μm-thick sagittal serial series from nasal to temporal regions (Figure 10) using a Leica cryostat CM-3050 (Leica Biosystems, Heidelberger, Germany). Tissue sections were collected on gelatin-coated slides (two sections per slide), air-dried, and stored at −30 °C until use. Analyses were performed taking into consideration the spatial organization of the retina all along the *x* axis in three regions, nasal (N), central (C) and temporal (T), and all along the *y* axis in three zones: superior (S), central (C) and Inferior (I). We obtained nine possible areas on which we performed the study: N-S, N-C, N-I, C-S, C-C, C-I, T-S, T-C and T-I (Figure 10).

**Figure 10 ijms-25-03115-f010:**
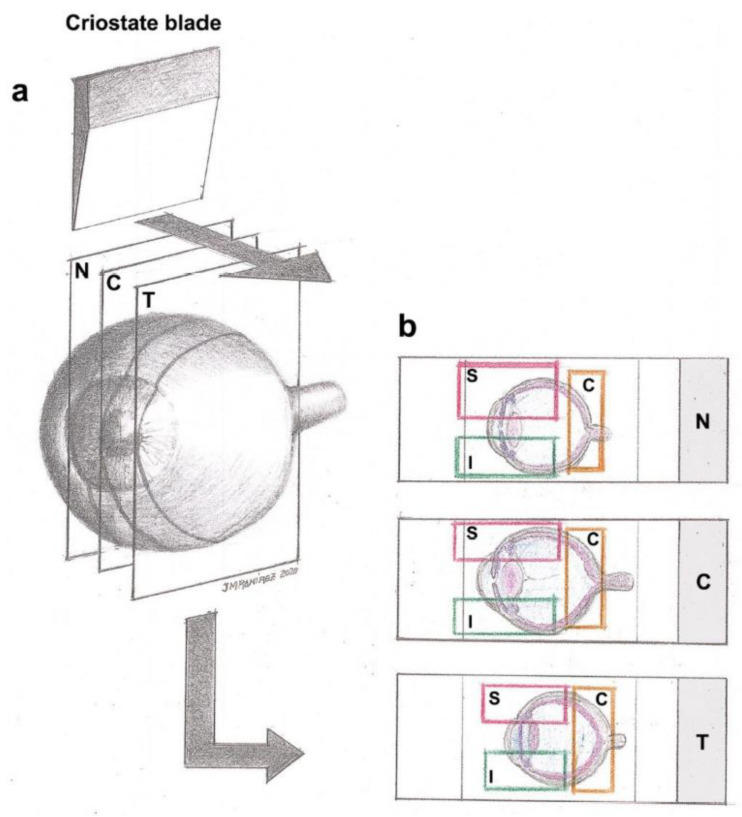
(**a**) Spatial distribution of the retina divided in nasal (N), central (C) and temporal (T) regions. Serial sagittal sections (16 μm-thick) taken from the nasal to temporal retina. (**b**) Tissue on gelatin-coated slides divided into regions (N, C or T) and superior (S), central (C) and inferior (I) retina zones, obtaining nine possible areas on which we performed the study: N-S, N-C, N-I, C-S, C-C, C-I, T-S, T-C and T-I. Reprinted from [8].

#### 4.6.1. Immunohistochemistry

A total of 5 animals per experimental group and per sacrifice day were randomly selected for immunohistochemical analyses. Slides containing the tissue sections were dried at room temperature for 10 min. Samples were then washed three times with immunohistochemistry buffer (IB) containing 0.5% Animal-Free Blocker^®^ and Diluent, R.T.U. (Vector Laboratories, Inc., Newark, CA, USA, Ref. SP-5035), and 0.3% Triton TM X-100 (Sigma-Aldrich, St. Louis, MO, USA, Ref. T8787) in 0.1 M PBS (Sigma-Aldrich, USA, Ref. P4417), pH 7.4. A solution of IB containing 0.5% of H_2_O_2_ was used to block the endogenous peroxidase for 15 min at room temperature. Subsequently, sections were incubated overnight at 4 °C, with the corresponding primary antibody diluted in IB with an additional 0.5% universal serum. The day after, slides were washed three times with IB and incubated for 2 h at room temperature with the secondary antibody diluted in IB (see Table 2 for details on the antibodies used). After incubation, sections were washed three times in IB and then incubated for 90 min at room temperature with an ABC Peroxidase Staining Kit (Thermo Scientific, Waltham, MA, USA, Ref. 32020), 1:250 dilution. The reaction product was revealed by incubating the sections with 0.45 mg/mL of 3, 3’-diaminobenzidine tetrahydrochloride hydrate (Sigma-Aldrich, USA, Ref. D5637) and 0.03% H_2_O_2_ in PBS. Finally, sections were coverslipped with Aquatex^®^ (Merck KGaE, Darmstadt, Germany, Ref. 1.08562.0050) mounting medium. Each immunohistochemical assay contained three slides (N, C and T) per experimental group, making a total of 18 slides plus a negative control slide with no primary antibody.

Immunostained slides were observed under a light microscope (Zeiss Axioplan Microscope, Oberkochen, Germany) linked to a high-resolution camera (Zeiss Axioplan 712 color, Germany), which was used for capturing the images. Image processing was performed in ZEN3.3 software, version 3.3.89.0000 (Carl Zeiss AG, Oberkochen, Germany). Magnification, light, shine and contrast conditions were kept constant during the capture process. Figures were prepared using Adobe Photoshop CS4 Extended 10.0 (Adobe Systems, San Jose, CA, USA).

#### 4.6.2. Quantitative Analysis

As previously explained in Figure 10, the retina was divided into nine different anatomical areas, which were taken into account for the analysis of different cell populations.

Retinal ganglion cells (RGCs)

Brn3a is a specific RGC marker [42]. In order to count the Brn3a-stained RGCs, the retinas were observed via light microscopy at 10× magnification (Zeiss Axioplan Microscope, Germany). Each of the 9 retinal zones was manually counted once by two different observers blind to the experimental groups, and only if the mean values had less than 15% interobserver differences were they used for the subsequent statistical analysis. Values obtained are expressed as positive Brn3a cells.

Macroglia: astrocytes and Müller cells

GFAP is a highly specific marker of macroglial activation in response to different forms of neuronal injury, and during microgliosis, astrocytes and Müller cells thicken and their GFAP expression increases [5]. The GFAP quantification of the different retinal zones was performed by micro photographing the immunostained slides via light microscopy at 20× (Zeiss Axioplan Microscope, Germany) with a high-resolution camera (Zeiss Axioplan 712 color, Germany). ZEN2 software (Carl Zeiss AG, Oberkochen, Germany) was used for image processing. Two different photos were taken of each of the 9 previously mentioned areas. We performed digital optical densitometry using ImageJ software, version 1.46r (Research Services Branch-NIH, Bethesda, MD, USA). The values of optic density (O.D.) are expressed in arbitrary units. In each of the photos, we analyzed the GFAP expression in each retinal layer independently: the ONFL-GCL, which was measured together as a unique layer, the inner plexiform layer (IPL), the inner nuclear layer (INL), the outer plexiform layer (OPL), the outer nuclear layer (ONL) and the photoreceptor layer (PL). We also analyzed the inner (ONFL-GCL, IPL, INL) and outer (OPL, ONL, PL) retina with the mean values of the corresponding layers, since this information allowed us to identify the activation level of Müller cells, which increase their GFAP expression towards the PL and pigmentary epithelium.

Microglial cells

Iba-1 is a specific microglial cell marker, whose expression is increased under microglial activation processes [17]. In order to count the Iba-1-stained microglial cells, retinas were observed via light microscopy at 10× magnification (Zeiss Axioplan Microscope, Germany). Each of the 9 retinal areas with its corresponding layers was counted once by two different observers blind to the experimental groups, and only if the mean values had less than 15% interobserver difference were they used for the subsequent statistical analysis. The retinal layers analyzed were ONFL-GCL, IPL, INL, OPL and ONL-PL (measured together as a unique layer). Under pathological conditions, there is a change in the pattern of the distribution of microglia in nuclear layers of the retina, and there are changes in their morphology from radial to migrating microglia. Considering the above, different microglial morphotypes were also analyzed, taking into consideration the layer where they were found and their orientation, establishing three morphotypes: radial microglia (RD), with processes all around; horizontal microglia (HZ), with their processes parallel to the retinal layers, occupying only one layer; and vertical microglia (VT), whose processes were disposed perpendicular to the retinal layers, going through one or more of them. Then, we established the percentage of each morphotype regarding the total Iba-1+ cells in each layer. The values obtained are expressed as positive Iba-1 cells.

### 4.7. Statistical Analysis

Two-way ANOVAs were performed for the two cohorts of animals employed, and sacrificed 3 days and 7 days after laser-induced OHT. The independent factors considered were the surgical intervention (S, with three levels: NAÏVE, OHT and CONTRA eyes) and the pharmacological treatment [Drug, D, with two levels: VH and CIL]. In the case of a statistically significant main effect of surgical intervention, results from Tukey post hoc comparisons were included. Moreover, whenever a statistically significant interaction was found, post hoc comparisons were performed by using the Tukey test. P-values below 0.05 were considered significant. A normal distribution and homoscedasticity were verified by using Shapiro–Wilk and Levene tests, respectively. Data are presented as the mean value and the standard deviation (SD) and were analyzed using SPSS software (SPPS 27, IBM, Armonk, NY, USA). Figures were prepared in GraphPad Prism version 8.0.2 (GraphPad Software, Inc., San Diego, CA, USA).

## 5. Conclusions

These promising results can lead to the first step in the development of a potential new treatment for glaucoma as part of a new anti-inflammatory and neuroprotective strategy. This treatment could not only be used as a nephroprotector, as has been described and postulated and for which clinical trials are underway, but also as an adjuvant for glaucoma IOP-reducing therapies, or even as a completely novel strategy for patients with normotensive glaucoma, for whom there is still no effective treatment to slow the vision loss caused by glaucoma. Further investigation of the molecular mechanisms underlying the results obtained would help to deepen the understanding of the action of cilastatin and, therefore, mark the beginning of the translation of the results to clinical applications.

## Figures and Tables

**Table 1 ijms-25-03115-t001:** Animal distribution in experimental groups.

		Day of Sacrifice			
		3 Days after Surgery		7 Days after Surgery	
Surgery	Pharmacological Treatment	Animals	Retinas	Animals	Retinas
NAÏVE (24)	NAÏVE + VH (12)	NAÏVE + VH (6) 1 dead animal prior to any manipulation.	NAÏVE + VH (10)	NAÏVE + VH (6) 1 dead animal prior to any manipulation.	NAÏVE + VH (10)
	NAÏVE + CIL (12)	NAÏVE + CIL (6)	NAÏVE + CIL (12)	NAÏVE + CIL (6)	NAÏVE + CIL (12)
OHT (48)	OHT + VH (24)	OHT + VH (12)	OHT + VH (12)	OHT + VH (12) 1 dead animal after surgery.	OHT + VH (11)
			CONTRA + VH (12)		CONTRA + VH (11)
	OHT + CIL (24)	OHT + CIL (12) 1 dead animal after surgery.	OHT + CIL (11)	OHT + CIL (12) 3 dead animals after surgery.	OHT + CIL (9)
			CONTRA +CIL (11)		CONTRA + CIL (9)

A total of 72 male Swiss mice were used in the present study. NAÏVE: control group without surgery; VH: vehicle; CIL: cilastatin; OHT: ocular hypertension; CONTRA: contralateral. The number of animals in the NAÏVE group was halved, since both eyes were used as independent samples for the NAÏVE group, but each eye of the animals exposed to the surgery was employed for two independent experimental groups: OHT (left eye) and CONTRA (right eye). The number of animals and retinas employed in each experimental group is presented in parenthesis.

**Table 2 ijms-25-03115-t002:** Antibodies and concentrations.

Cell Population	Primary Antibody	Secondary Antibody
RGCs	Mouse monoclonal anti-Brn3a (Ref. MAB1585, Millipore, Burlington, MA, USA) [1:600]	Goat-Anti-mouse IgG(H + L) Biotinylated (Ref. BA9200, Vector Laboratories, Newark, CA, USA) [1:200]
Macroglia: Müller cells and astrocytes	Rabbit-Anti GFAP (polyclonal) (Ref. Z0334, Dako, Agilent, Santa Clara, CA, USA) [1:800]	Goat-Anti-rabbit IgG (H + L) Biotinylated (Ref. 31820, Thermo Scientific, Waltham, MA, USA) [1:200]
Microglia	Rabbit (polyclonal) anti Iba-1 (Ref. 019-19471, Wako Chemicals, Richmond, VA, USA) [1:500]	

RGCs: retinal ganglion cells; GFAP: glial fibrillary acidic protein; Iba-1: ionized calcium-binding adapter molecule 1. For each antibody, the commercial distributor is indicated, together with the specific antibody reference and the concentration employed in the present study.

## Data Availability

The data presented in this study are available on request from the corresponding author. The data are not publicly available due to patentabilty concerns.

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
