# Peer review of "Cilastatin as a Potential Anti-Inflammatory and Neuroprotective Treatment in the Management of Glaucoma"

_ijms, 2024, doi:10.3390/ijms25063115_

Round 1

Reviewer 1 Report

Comments and Suggestions for Authors

The authors have presented significant research by deciphering the potential of cilastatin as a potential anti-inflammatory and neuroprotective 2 strategy in the management of glaucoma. They have emphasized 38 effective strategies to reduce OHT-associated damage in the retina of experimental mice. This would be significant research in the field of glaucoma management. 

This study is novel and highly relevant in the field of medicine and research. It has addressed the gaps reported in the management of glaucoma disease in an experimental mouse model based on the unilateral (left) laser- 28 induced ocular-hypertension which has not been addressed in previous studies. 

These experimental findings comprehend the first step in the development of a potential new treatment for glaucoma 818 with a new anti-inflammatory approach that could be used not only as a neuroprotector, 819 as has been described and postulated and for which clinical trials are underway (Eu- 820 draCT Number: 2022-001417-39), but also as an adjuvant for glaucoma IOP-reducing 821 therapies, that could cover their partial lack of effectiveness; or even as a completely novel 822 strategy for the patients with normotensive glaucoma that still do not have an effective 823 treatment in the withholding of the vision loss caused by glaucoma. 

I would only recommend the authors kindly rewrite the conclusion section and remove citations from the conclusion section. The methodology is very well elaborated highlighting the details of experiments needed to perform. However, the authors are suggested to improve the quality of graphs.

The conclusions are highly consistent with the evidence and arguments presented and they addressed the significance of cilastatin as a potential anti-inflammatory and neuroprotective 2 strategy in the management of glaucoma. However, I have suggested the authors remove the reference from the conclusion section. Also, try to add a futuristic approach to the research that would help future researchers to design their study.

All the reported references are highly appropriate.

Altogether, I would recommend the publication of this manuscript.

Comments on the Quality of English Language

Minor editing is needed

Reviewer 2 Report

Comments and Suggestions for Authors

An interesting study, it just needs to be presented better.

Introduction: Fine

Materials - messy and missing details.  There needs to be enough information for the study to be replicated.

line 631 - how many mice on total injected with CIL?  72 total.  24 with CIL, 24 with saline, the rest had nothing?

line 634-5: so who got 6 and who got 10?

Line 642: - 48 mice has surgery?

line 656: is the beginning of the experiment the same as 2 days before treatment?  Or upon surgery? What was intervention?

section 4.5 - written in a very confusing manner.  Perhaps a table to show who has what done to them.  e.g.: 48 surgery, two groups of 24 (3 day and 7 day) - either further split into 2 groups of 12: for CIL and Saline.  In a table, you can clearly show which one died before treatment completed too.

section 4.6: this is "Tissue preparation" not IHC analysis!

line 722 - subscript number in peroxide!

Results - over written and very confusing.  Since you have such detailed tables, no need to re-describe them.

E.g. Results of IOP measurement are shown in figure 1.  In general the operation resulted in an increase in IOP, which was not mitigated by the presence of CIL.  There appeared to be little impact on the CONTRA side. 

E.g.: Results of Brn3a+ shown in figures 2 and 3.  In general there was a post-operative decrease in Brn3+, which was partially offset by the application of CIL in both operated and CONTRA sides. 

E.g.: Results of GFAP+ shown in figure 4.  In general, there was a post-surgical increase in GFAP+, which was suppressed by the presence of CIL in both operative and CONTRA sides.

etc...

If you present all that data in tabular form, there is no need to repeat it, especially when the aim was simply to show if CIL had an impact.  Easier to see the data than battle through impenetrable prose.

As for figure legends: the word in there is "Location" not "Evolution".  This is NOT an evolutionary study, it is an experimental study, please use the correct terminology.  Figure 1 - remove the word "Evolution", no need to replace with "Location".

Figure 3, 5 and 8 are underwhelming.  They are not labelled and look like black lines on a page.  IF there is anything to show, please label clearly.  Just because it was done does NOT mean you have to show it.  Are they all the same?  Are some the same?  Can you show "type" pictures when they look the same?  Scale bar MUST be on every pictures, not just the bottom corner.

Figure 6 - where did the notion of inner and outer come from?  Not mentioned in the materials section.  

Why are there statistical tables complementing the data from the graphs?  Supplementary data?  Does it bring anything extra?

Discussion - unfocussed.

The question posed at the end of the introduction was "what is the effect of CIL on surgically altered animals".  This manuscript examined the effect of surgery and CIL on OPL and a variety of cellular factors.  Surgery impacted OPL, but CIL had no effect.  That was not the case with the cellular components. 

Set up the rest of the discussion to explore each experiment (OPL, Brn3+ etc) first explaining the effect of the surgery (as done) and THEN explaining the effect of CIL (not in a tiny paragraph at the end).  Each section with it's own subheading to make it easier to follow.

Conclusion - do NOT cite new work in the conclusion.  The point of the conclusion is to put together the overall findings (CIL not impact OPL but does impact cellular activity), significance and what needs to be be done next,

Reviewer 3 Report

Comments and Suggestions for Authors

The manuscript submitted for review presents the results of a study evaluating the effect of cisplatin on the reduction of laser-induced oculer-hypertension in an animal model. In addition, the effects of cisplatin use on Retinal Ganglion Cells (RGCs) survival, and inflammatory responses of macroglial and microglial cells were evaluated. After reading the manuscript, please answer the following questions or correct certain passages.

1. Why are two "naive" groups (VH and CIL) present in some of the studies, while in others there is only one such group?

2. too many results like "F = ...... ; p < 0.001". The value of the F test statistic is not very informative. Moreover, the p-value of an ANOVA (two-factor, three-factor, or mixed models) also tells the reader little - the results of post-hoc tests are the most important.

3 The tables show the results of the two-factor ANOVA. In many situations, the authors provide the results of the main effects and interactions between them. Providing the results of the main effects is of little importance in such studies - the most important are the interactions between these factors. In general, these tables are completely unnecessary in the manuscript, there are many other results in it.

4. Figure 1: "Two-way ANOVA per time point: a p<0.05, aaa p<0.005, aaaa p<0.001 vs NAÏVE group; αααα p<0.001 vs OHT group." Are the reported values the results of a post-hoc test? Are these results for one main effect or interaction?

5. Figure 2: "Two-way ANOVA: general effect of the surgery: a p<0.05, aa p<0.01, aaa p<0.005, aaaa p<0.001 vs NAÏVE group; α p<0.05, αα p<0.01, ααα p<0.005, ααα p<0.001 vs OHT group; general effect of CIL treatment: b p<0.05." Are these the results of a post-hoc test? I have a similar question for Figures 4, 6, 9.

6. Figure 2: "In case of a significant interaction between factors, Tukey post-hoc comparisons: *p<0.05, **p<0.01, ***p<0.005, ****p<0.001 vs NAÏVE+VH; # p<0.05, ## p<0.01, ### p<0.005, #### p<0.001 vs OHT+VH." 

Were only post-hoc test results that were statistically significant given? I have a similar question for figures 4, 6, 9.

7. Do the authors feel the paper had any limitations or critical points?

8.      Lines 595-605: The authors refer to their previous work in the last paragraph. In my opinion, this is not the right place for this type of practice. In addition, the authors mention two previous studies, and one is cited. This paragraph should summarize the entire discussion focusing primarily on the results currently presented.

Round 2

Reviewer 3 Report

Comments and Suggestions for Authors

After reading the revised manuscript and the author's responses, I would very much appreciate if you could answer the following questions or correct certain sections.

1 The authors continue, very often, to report main effects results, which are of little relevance in a situation where the primary research objective is to evaluate CIL activity (vs VH) [one factor] with a simultaneous division into OHT intervention (vs CONTA vs NAIVE) [the other factor]. The main effects analysis in such a situation is misleading, e.g. the OHT vs. CONTRA comparison is a comparison of all OHT animals (simultaneously for those that had CIL or VH administered) vs. all CONTRA animals (simultaneously for those that had CIL or VH administered). I would have loved to see the results without all the symbols indicating the main effects in the graphs. Unfortunately, the authors did not address this comment in the first round of reviews.

2. lines 720-722: "In case of statistically significant interactions post-hoc comparisons were performed by using the Tukey test. P-values below 0.05 were considered significant." How was statistical significance assessed for main effects? Did you perform any post-hoc tests here?

3 Some sentences erroneously imply a reference to the authors' previous works, where works by other authors are cited. For example: "We also propose the signaling through the optic nerve of the OHT eye [32] explaining the higher microglial activation in the nasal and central areas of the CONTRA eye."

Round 3

Reviewer 3 Report

Comments and Suggestions for Authors

The changes made are sufficient. In my opinion, the article can be accepted for publication.